# Lifestyle behaviours and associated factors among people with type 2 diabetes attending a diabetes clinic in Ningbo, China: A cross-sectional study

**Naomi Carter** [1], **Jialin Li**[2], **Miao Xu**[2], **Li Li**[2], **Shengnan Xu**[2], **Xuelan Fan**[2], **Shuyan Zhu**[2], **Prit Chahal**[3], **Kaushik Chattopadhyay** [1,4]*

**1** Lifespan and Population Health, School of Medicine, University of Nottingham, Nottingham, United Kingdom, **2** Department of Endocrinology and Metabolism, The First Affiliated Hospital of Ningbo University, Ningbo, People's Republic of China, **3** Health Education England, Leeds, United Kingdom, **4** The Nottingham Centre for Evidence-Based Healthcare, A JBI Centre of Excellence, Nottingham, United Kingdom

* Kaushik.Chattopadhyay@nottingham.ac.uk

**Data Availability Statement:** All relevant data are within the manuscript and its Supporting Information files.

## Abstract

The burden of type 2 diabetes (T2DM) in China is significant and growing, and this is reflected in high rates of T2DM in the city of Ningbo, China. Consequent impacts on morbidity, mortality, healthcare expenditure, and health-related quality of life, make this a problem of the utmost importance to address. One way to improve T2DM outcomes is to address lifestyle behaviours that may affect prognosis and complications, such as physical activity levels, dietary habits, smoking status, and alcohol intake. A cross-sectional survey was undertaken to describe the prevalence of being physically active, having a healthy diet, currently smoking, and currently drinking alcohol among people living with T2DM attending a diabetes clinic in Ningbo, China. Regression analysis was used to determine the factors associated with these lifestyle behaviours. We found a high prevalence of a healthy diet (97.8%, 95% CI 96.5–98.7%). Prevalence of being physically active (83.4%, 95% CI 80.6–85.9%), smoking (21.6%, 95% CI 18.8–24.6%), and alcohol drinking (32.9%. 95% CI 29.6–36.2%) appeared in keeping with those of the general population. Marked associations were demonstrated between male sex and smoking (OR 41.1, 95% CI 16.2–139.0), and male sex and alcohol drinking (OR 4.00, 95% CI 2.62–6.20). Correlation between lifestyle factors was demonstrated including between alcohol drinking and smoking, and between physical activity and reduced smoking. General diabetes self-management education programmes that address multiple lifestyle risk factors simultaneously may be beneficial in this population. Specific interventions targeting smoking cessation and reduction in alcohol drinking may be of benefit to men living with T2DM attending a diabetes clinic in Ningbo.

**Funding:** The study was funded by the Nottingham China Health Institute, China, and the Major Science and Technology Projects for Health of Zhejiang Province (Grant No. WKJ-ZJ-2216). NC was an In-Practice Fellow supported by the UK's Department of Health and Social Care and the National Institute for Health Research (NIHR301000). The views expressed are those of the authors and not necessarily those of the NHS, the NIHR or the Department of Health and Social Care. https://www.nihr.ac.uk/ The funders had no role in study design, data collection and analysis, decision to publish, or preparation of the manuscript.

**Competing interests:** The authors have declared that no competing interests exist.

# Introduction

Type 2 diabetes (T2DM) prevalence is growing rapidly in China, causing a significant and increasing burden of morbidity and mortality [1–3]. In 2021, the age-adjusted prevalence of diabetes in people aged 20–79 in China reached 10.6%, and an estimated 51.7% of those with diabetes were thought to be undiagnosed [4]. Diabetes is thought to be directly responsible for 3.9% of deaths in those under 60 [4]. Expenditure on diabetes-related health costs was estimated at 165,304 million US dollars in 2021, and this is projected to continue growing [4].

Ningbo is an economically developed city in the eastern, coastal province of Zhejiang. The prevalence of T2DM in Ningbo exceeds national averages, with over 20% of the over-40s affected [5]. Consequences include high rates of micro- and macro-vascular complications amongst those living with T2DM, as well as negative health-related quality-of-life impacts; particularly in the domains of depression, anxiety, and pain [6,7]. Rates of comorbidities and consequent polypharmacy are high amongst this population living with T2DM, further contributing to morbidity [8,9].

Lifestyle behaviours such as physical activity levels, dietary habits, smoking status, and alcohol intake can impact complication rates and prognosis in T2DM [10–15]. Lifestyle behaviours are also heavily contextual, intertwined with how people live their lives, and thus warrant contextual assessment. Through a better understanding of the distribution of various lifestyle behaviours and the factors associated with them, it may be possible to efficiently target lifestyle interventions towards key areas of need. Therefore, we aimed to describe the prevalence of being physically active, having a healthy diet, currently smoking, and currently drinking alcohol among people living with T2DM attending a diabetes clinic in Ningbo, China; and to determine the factors associated with these lifestyle behaviours.

# Methods

## Study design and setting

This cross-sectional study was carried out at the diabetes outpatient clinic of the First Affiliated Hospital of Ningbo University, Ningbo, China. Data were collected from 1st June 2020 to 31st May 2021. The study site was chosen based on an existing collaborative partnership between the University of Nottingham campus in Ningbo, China, and the First Affiliated Hospital of Ningbo University, Ningbo, China; which is a tertiary care hospital including a diabetes centre with a team of qualified and experienced diabetes experts [16]. As well as providing specialist health and medical services, the First Affiliated Hospital of Ningbo University is involved in medical research and education [6].

## Sampling strategy

At least 784 participants were needed based on a 95% confidence level, a 3.5% margin of error, and a conservative population proportion of 50% [17]. During the time period of the study, eligible clinic patients were approached consecutively by the physicians and invited to participate. No incentives to participate (financial or otherwise) were provided.

## Study population and eligibility criteria

The study included adult ($\geq$ 18 years) patients with T2DM, with written informed consent to participate. The diagnosis of T2DM was based on the Chinese guideline, which was in keeping with World Health Organization recommendations for diagnosis [18].

## Data collection tool and procedures

A survey was conducted with a quantitative questionnaire, administered in Mandarin. The self-reported non-standardised questions were developed and pretested among six patients with T2DM (not included in this study). Data were collected and entered by a team of five trained nurses. Another staff member was responsible for checking the data quality. Data on the following variables were collected: age (18–39 years, 40–59 years, or $\geq$ 60 years), sex (female or male), education level (no qualification, class 1–6, class 7–12, or college/university), occupation (manual worker, non-manual worker, or never worked/retired), residence (urban or rural as defined based on the 'hukou' registration system), marital status (married or single/widowed/divorced), health insurance (yes or no), duration of T2DM ($\leq$1 year, >1–5 years, >5–10 years, or >10 years), and family history of T2DM (no or yes; T2DM in any parent or sibling).

Data were also collected on biochemical, anthropometric, and physiological parameters. The anthropometric and physiological parameters were measured twice per participant, and the average of these two readings was used in the analysis. Glycosylated haemoglobin (HbA1c) was measured using venous blood sampling and the high-performance liquid chromatographic method (D-10 Hemoglobin Analyzer, Bio-Rad). Controlled T2DM was defined as HbA1c < 7% in accordance with the Chinese T2DM guideline [18]. Body weight (kilograms [kg] to one decimal place), height (metres [m] to two decimal places), and waist circumference (centimetres [cm] to the nearest half cm) were measured. Body weight and height were measured with light clothes and without shoes in the standing position using a calibrated automatic digital weight and height scale (HNH-318, Omron, Japan). Body mass index (BMI) was calculated as body weight (kg) divided by height (m) squared. Participants were categorised as 'not overweight' (BMI < 23 kg/m$^2$), 'overweight' (BMI $\geq$ 23.0 < 27.5 kg/m$^2$), or 'obese' (BMI $\geq$ 27.5 kg/m$^2$) based on reference ranges specific to the Chinese population [19]. Waist circumference was measured using a 150 cm medical tape at the midpoint between the lower rib and iliac crest. Abdominal obesity was defined as a waist circumference of >90 cm in males or >80 cm in females based on reference ranges specific to the Chinese population [19]. Blood pressure was measured on the day of the visit in a seated position using a calibrated, automated sphygmomanometer (HBP-1100U, Omron, Japan), and hypertension was defined as $\geq$140/90 mmHg in keeping with Chinese guidelines [20].

The outcomes in the present study were the prevalence of four lifestyle behaviours including two positive behaviours (being physically active and having a healthy diet) and two negative behaviours (being a current smoker and being a current alcohol drinker). This paper focused on current rather than retrospective healthy and unhealthy lifestyle behaviours based on the rationale that these are potentially modifiable, whereas past lifestyle behaviours are not. Data on physical activity were obtained using the validated IPAQ questionnaire (International Physical Activity Questionnaire—short form), available in Mandarin [21]. Being physically active was defined as currently meeting the criteria for the moderate or high physical activity IPAQ categories [21]. Data on diet were obtained using a modified version of the UK Diabetes and Diet Questionnaire (UKDDQ) [22]. The questions were adapted to be relevant to dietary norms in China, and the questionnaire was translated into Mandarin and pre-tested (see above). A healthy diet was defined as currently having more than 50% A + B answers on the UKDDQ which correspond to 'healthy' dietary choices [22]. Data on current smoking and alcohol drinking status were obtained through self-reporting via the aforementioned quantitative questionnaire.

Participants were assigned a participant identification number which the authors had access to during and after data collection.

### Ethics approval

Ethics approval was received from the Research Ethics Committee of the First Affiliated Hospital of Ningbo University, Ningbo, China (ref. 2019-R057). The participant information sheet and consent form were available in Mandarin. The study objective was explained to all the eligible participants, and written informed consent was obtained from those interested in participating. Additional information regarding the ethical, cultural, and scientific considerations specific to inclusivity in global research is included in the (Appendix 1: S1 Checklist)

### Statistical analyses

Data were entered into R (version 4.1.2, R Core Development Team) for statistical analysis. During data cleaning, data were screened for implausible values (data cleaning completed by NC), and these were deleted where found and treated as missing. If the implausible value was one of a pair of repeated measures, both measures were deleted and treated as missing. Descriptive statistics were calculated including median and interquartile range (IQR) for the non-normally distributed continuous variable of age, and numbers and percentages for categorical variables (all other variables). Prevalence was calculated as the number of participants with the lifestyle behaviour as a proportion of all participants, with Clopper-Pearson 95% confidence intervals (CIs) calculated using the PropCIs package in R. Univariate and multivariable logistic regressions were undertaken to measure the unadjusted and adjusted associations respectively between the independent variables and the four outcomes (being physically active, having a healthy diet, current smoking, and current alcohol drinking), using the glm() function in R. This produced odds ratios (ORs) with 95% CIs. Independent variables were selected for inclusion in the multivariable model using a significance cut-off of $p \leq 0.2$ in the univariate analysis. The overall significance cut-off was set at $p \leq 0.05$. Missing data were excluded listwise.

A sensitivity analysis was conducted using waist-to-height ratio (WHtR) $\geq 0.5$ to define abdominal obesity rather than the previously specified waist circumference thresholds [23].

## Results

A total of 806 patients with T2DM participated in the study. The median age among participants was 56 years (IQR 47–64), and 59.6% (480/806) of participants were male. Further demographic, clinical and lifestyle characteristics are provided in Table 1. Most participants reported being physically active (672/806, 83.4%, 95% CI 80.6–85.9%) and having a healthy diet (788/806, 97.8%. 95% CI 96.5–98.7%). Fewer participants were current smokers (174/806, 21.6%, 95% CI 18.8–24.6%) and current alcohol drinkers (265/806, 32.9%. 95% CI 29.6–36.2%).

In multivariable regression analysis, living in a rural setting (versus urban) was associated with reduced odds of being physically active (OR 0.59, 95% CI 0.39–0.88) [Table 2]. Being single/widowed/divorced (versus married) was associated with reduced odds of having a healthy diet (OR 0.23, 95% CI 0.07–0.96). Male sex (versus female) was associated with a marked increase in the odds of being a current smoker (OR 41.1, 95% CI 16.2–139.0). Being an alcohol drinker was also associated with increased odds of being a smoker (OR 1.85, 95% CI 1.22–2.82). Factors which were associated with reduced odds of smoking included college/university education (versus no qualification) [OR 0.19, 95% CI 0.06–0.65], being single/widowed/divorced (versus married) [OR 0.54, 95% CI 0.09–0.80], and being physically active (versus not physically active) [OR 0.56, 95% CI 0.33–0.96]. Several factors were associated with increased odds of being an alcohol drinker including male sex (versus female) [OR 4.00, 95% CI 2.62–6.20], being a non-manual worker (versus manual) [OR 1.69, 95% CI 1.01–2.83], being a

**Table 1. Participant characteristics and factors associated with lifestyle behaviours assessed by univariate regression analysis.**

| Participant characteristics (Total participants = 806) | | | Physically active | | Healthy diet | |
|---|---|---|---|---|---|---|
| | | | N (%) | Unadjusted OR (95% CI) [p-value] | N (%) | Unadjusted OR (95% CI) [p-value] |
| Age (N [%]) | 18–39 years | 115 (14.3) | 93 (80.9) | Ref | 109 (94.8) | Ref |
| | 40–59 years | 379 (47.0) | 316 (83.4) | 1.19 (0.68–2.01) [0.533] | 368 (97.1) | **2.25 (0.74–6.38) [0.132]** |
| | ≥ 60 years | 309 (38.3) | 260 (84.1) | 1.26 (0.71–2.17) [0.423] | 308 (99.7) | **17.00 (2.85–322.00) [0.009]** |
| | Missing | 3 (0.4) | 3 (100.0) | N/A | 3 (100.0) | N/A |
| Sex (N [%]) | Female | 325 (40.3) | 269 (82.8) | Ref | 321 (98.8) | Ref |
| | Male | 480 (59.6) | 402 (83.8) | 1.07 (0.73–1.56) [0.714] | 466 (97.1) | 0.48 (0.13–1.40) [0.212] |
| | Missing | 1 (0.1) | 1 (100.0) | N/A | 1 (100.0) | N/A |
| Education level (N [%]) | No qualification | 64 (7.9) | 54 (84.4) | Ref | 64 (100.0) | N/A* |
| | Class 1–6 | 177 (22.0) | 141 (79.7) | 0.73 (0.32–1.52) [0.412] | 174 (98.3) | Ref |
| | Class 7–12 | 423 (52.5) | 357 (84.4) | 1.00 (0.46–2.00) [0.996] | 410 (96.9) | 0.59 (0.13–1.88 [0.417] |
| | College/university | 142 (17.6) | 120 (84.5) | 1.01 (0.43–2.23) [0.981] | 140 (98.6) | 2.41 (0.31–49.1) [0.448] |
| Occupation (N [%]) | Manual worker | 153 (19.0) | 123 (80.4) | Ref | 148 (96.7) | Ref |
| | Non-manual worker | 287 (35.6) | 233 (81.2) | 1.05 (0.64–1.72) [0.840] | 276 (96.2) | 1.04 (0.31–3.06) [0.950] |
| | Never worked/retired | 366 (45.4) | 316 (86.3) | **1.54 (0.93–2.52) [0.089]** | 364 (99.5) | **6.15 (1.31–43.3) [0.031]** |
| Residence (N [%]) | Urban | 482 (59.8) | 417 (86.5) | Ref | 470 (97.5) | Ref |
| | Rural | 323 (40.1) | 254 (78.6) | **0.57 (0.40–0.83) [0.004]** | 317 (98.1) | 1.12 (0.41–3.33) [p = 0.822] |
| | Missing | 1 (0.1) | 1 (100.0) | N/A | 1 (100.0) | N/A |
| Marital status (N [%]) | Married | 763 (94.7) | 637 (83.5) | Ref | 749 (98.2) | Ref |
| | Single/widowed/divorced | 43 (5.3) | 35 (81.4) | 0.87 (0.41–2.05) [0.720] | 39 (90.7) | **0.16 (0.05–0.58) [0.002]** |
| Health insurance (N [%]) | Yes | 772 (95.8) | 648 (83.9) | Ref | 755 (97.8) | Ref |
| | No | 34 (4.2) | 24 (70.6) | **0.46 (0.22–1.03) [0.045]** | 33 (97.1) | 0.66 (0.13–12.00) [0.687] |
| Duration of T2DM (N [%]) | ≤1 years | 186 (23.1) | 152 (81.7) | Ref | 181 (97.3) | Ref |
| | >1–5 years | 190 (23.6) | 154 (81.1) | 0.96 (0.57–1.61) [0.868] | 188 (98.9) | 2.60 (0.55–18.30) [0.258] |
| | >5–10 years | 156 (19.4) | 132 (84.6) | 1.23 (0.70–2.20) [0.478] | 150 (96.2) | 0.83 (0.23–3.03) [0.770] |
| | >10 years | 272 (23.7) | 232 (85.3) | 1.30 (0.78–2.14) [0.308] | 267 (98.2) | 1.84 (0.48–7.54) [0.367] |
| | Missing | 2 (0.2) | 2 (100.0) | N/A | 2 (100.0) | N/A |
| Family history of T2DM (N [%]) | No | 435 (54.0) | 356 (81.8) | Ref | 425 (97.7) | Ref |
| | Yes | 370 (45.9) | 315 (85.1) | 1.27 (0.88–1.86) [0.212] | 362 (97.8) | 0.85 (0.31–2.34) [0.751] |
| | Missing | 1 (0.1) | 1 (100.0) | N/A | 1 (100) | N/A |
| Controlled T2DM (N [%]) | Yes | 272 (33.7) | 226 (83.1) | Ref | 268 (98.5) | Ref |
| | No | 517 (64.1) | 433 (83.8) | 1.05 (0.70–1.55) [0.811] | 503 (97.3) | **0.42 (0.10–1.36) [0.195]** |
| | Missing | 17 (2.1) | 13 (76.5) | N/A | 17 (100) | N/A |
| Body mass index (BMI) (N [%]) | Not overweight | 268 (33.3) | 219 (81.7) | Ref | 262 (97.8) | Ref |
| | Overweight | 380 (47.1) | 320 (84.2) | 1.19 (0.79–1.80) [0.404] | 373 (98.2) | 1.19 (0.34–3.98) [0.780] |
| | Obese | 149 (18.5) | 124 (83.2) | 1.11 (0.66–1.91) [0.700] | 144 (96.6) | 0.55 (0.15–2.01) [0.350] |
| | Missing | 9 (1.1) | 9 (100.0) | N/A | 9 (100.0) | N/A |
| Abdominal obesity (N [%]) | No | 314 (39.0) | 264 (84.1) | Ref | 306 (97.5) | Ref |
| | Yes | 479 (59.4) | 395 (82.5) | 0.89 (0.64–1.30) [0.553] | 469 (97.9) | 1.19 (0.42–3.23) [0.730] |
| | Missing | 13 (1.6) | 13 (100.0) | N/A | 13 (100.0) | N/A |
| Hypertension (N [%]) | No | 613 (76.1) | 504 (82.2) | Ref | 598 (97.6) | Ref |
| | Yes | 189 (23.4) | 164 (86.8) | **1.42 (0.90–2.31) [0.144]** | 186 (98.4) | 1.35 (0.43–5.92) [0.644] |
| | Missing | 4 (0.5) | 4 (100.0) | N/A | 4 (100.0) | N/A |
| Physically active (N [%]) | No | 134 (16.6) | | | 129 (96.3) | Ref |
| | Yes | 672 (83.4) | | | 659 (98.1) | **2.32 (0.72–6.50) [0.124]** |
| Healthy diet (N [%]) | No | 16 (2.0) | 11 (68.8) | Ref | | |
| | Yes | 788 (97.8) | 659 (83.6) | **2.32 (0.72–6.50) [0.124]** | | |
| | Missing | 2 (0.2) | 2 (100) | N/A | | |
| Smoking (N [%]) | No | 628 (77.9) | 533 (84.9) | Ref | 615 (97.9) | Ref |
| | Yes | 174 (21.6) | 135 (77.6) | **0.62 (0.41–0.95) [0.024]** | 169 (97.1) | 0.82 (0.28–2.98) [0.741] |
| | Missing | 4 (0.5) | 4 (100.0) | N/A | 4 (100.0) | N/A |
| Alcohol drinking (N [%]) | No | 538 (66.7) | 445 (82.7) | Ref | 527 (98.0) | Ref |
| | Yes | 265 (32.9) | 224 (84.5) | 1.14 (0.77–1.72) [0.517] | 258 (97.4) | 0.82 (0.30–2.42) [0.697] |
| | Missing | 3 (0.4) | 3 (100.0) | N/A | 3 (100.0) | N/A |

(*Continued*)

**Table 1.** (Continued)

| Participant characteristics (Total participants = 806) | | | Physically active | | Healthy diet | |
|---|---|---|---|---|---|---|
| | | | N (%) | Unadjusted OR (95% CI) [p-value] | N (%) | Unadjusted OR (95% CI) [p-value] |
| Age (N [%]) | 18–39 years | 115 (14.3) | 35 (30.2) | Ref | 34 (29.6) | Ref |
| | 40–59 years | 379 (47.0) | 99 (26.1) | 0.82 (0.52–1.30) [0.388] | 151 (39.8) | **1.60 (1.03–2.53) [0.041]** |
| | ≥ 60 years | 309 (38.3) | 38 (12.3) | **0.32 (0.19–0.54) [<0.001]** | 78 (25.2) | 0.80 (0.50–1.30) [0.370] |
| | Missing | 3 (0.4) | 2 (66.7) | N/A | 2 (100.0) | N/A |
| Sex (N [%]) | Female | 325 (40.3) | 4 (1.2) | Ref | 48 (14.8) | Ref |
| | Male | 480 (59.6) | 170 (35.4) | **44.6 (18.6–146.0) [<0.001]** | 217 (45.2) | **4.82 (3.40–6.93) [<0.001]** |
| | Missing | 1 (0.1) | 0 (0) | N/A | 0 (0) | N/A |
| Education level (N [%]) | No qualification | 64 (7.9) | 8 (12.5) | Ref | 11 (17.2) | Ref |
| | Class 1–6 | 177 (22.0) | 21 (11.9) | 0.94 (0.41–2.37) [0.893] | 50 (28.2) | 1.90 (0.94–4.09) [0.084] |
| | Class 7–12 | 423 (52.5) | 117 (27.7) | **2.71 (1.32–6.31) [0.011]** | 150 (35.5) | **2.68 (1.41–5.54) [0.005]** |
| | College/university | 142 (17.6) | 28 (19.7) | 1.72 (0.77–4.26) [0.211] | 54 (38.0) | **2.96 (1.46–6.41) [0.004]** |
| Occupation (N [%]) | Manual worker | 153 (19.0) | 41 (26.8) | Ref | 45 (29.4) | Ref |
| | Non-manual worker | 287 (35.6) | 97 (33.8) | 1.40 (0.91–2.17) [0.131] | 127 (44.3) | **1.91 (1.26–2.92) [0.002]** |
| | Never worked/retired | 366 (45.4) | 36 (9.8) | **0.30 (0.18–0.49) [<0.001]** | 93 (25.4) | 0.81 (0.53–1.24) [0.326] |
| Residence (N [%]) | Urban | 482 (59.8) | 107 (22.2) | Ref | 172 (35.7) | Ref |
| | Rural | 323 (40.1) | 66 (20.4) | 0.90 (0.63–1.26) [0.535] | 92 (28.5) | **0.72 (0.53–0.97) [0.032]** |
| | Missing | 1 (0.1) | 1 (100.0) | N/A | 1 (100.0) | N/A |
| Marital status (N [%]) | Married | 763 (94.7) | 169 (22.1) | Ref | 257 33.7) | Ref |
| | Single/widowed/divorced | 43 (5.3) | 5 (11.6) | **0.46 (0.16–1.08) [0.108]** | 8 (18.6) | **0.45 (0.19–0.93) [0.044]** |
| Health insurance (N [%]) | Yes | 772 (95.8) | 168 (21.8) | Ref | 258 (33.4) | Ref |
| | No | 34 (4.2) | 6 (17.6) | 0.76 (0.28–1.76) [0.559] | 7 (20.6) | **0.51 (0.20–1.13) [0.122]** |
| Duration of T2DM (N [%]) | ≤1 years | 186 (23.1) | 55 (29.6) | Ref | 72 (38.7) | Ref |
| | >1–5 years | 190 (23.6) | 46 (24.2) | 0.75 (0.48–1.19) [0.228] | 56 (29.5) | **0.66 (0.43–1.01) [0.050]** |
| | >5–10 years | 156 (19.4) | 36 (23.1) | **0.70 (0.43–1.14) [0.158]** | 49 (31.4) | **0.71 (0.45–1.11) [0.139]** |
| | >10 years | 272 (23.7) | 37 (13.6) | **0.37 (0.23–0.59) [<0.001]** | 87 (32.0) | **0.73 (0.50–1.08) [0.117]** |
| | Missing | 2 (0.2) | 0 (0) | N/A | 1 (50.0) | N/A |
| Family history of T2DM (N [%]) | No | 435 (54.0) | 101 (23.2) | Ref | 127 (29.2) | Ref |
| | Yes | 370 (45.9) | 73 (19.7) | 0.82 (0.58–1.15) [0.248] | 138 (37.3) | **1.46 (1.09–1.97) [0.012]** |
| | Missing | 1 (0.1) | 0 (0.0) | N/A | 0 (0.0) | N/A |
| Controlled T2DM (N [%]) | Yes | 272 (33.7) | 52 (19.1)23.0) | Ref | 101 (37.1) | Ref |
| | No | 517 (64.1) | 119 (23.0) | 1.27 (0.89–1.84) [0.201] | 160 (30.9) | **0.76 (0.56–1.03) [0.080]** |
| | Missing | 17 (2.1) | 3 (17.6) | N/A | 4 (23.5) | N/A |
| Body mass index (BMI) (N [%]) | Not overweight | 268 (33.3) | 52 (19.4) | Ref | 69 (25.7) | Ref |
| | Overweight | 380 (47.1) | 87 (22.9) | 1.24 (0.84–1.83) [0.282] | 142 (37.4) | **1.72 (1.22–2.43) [0.003]** |
| | Obese | 149 (18.5) | 34 (22.8) | 1.23 (0.75–2.00) [0.400] | 50 (33.6) | **1.46 (0.94–2.27) [0.087]** |
| | Missing | 9 (1.1) | 1 (11.1) | N/A | 4 (44.4) | N/A |
| Abdominal obesity (N [%]) | No | 314 (39.0) | 81 (25.8) | Ref | 101 (32.2) | Ref |
| | Yes | 479 (59.4) | 92 (19.2) | **0.68 (0.48–0.95) [0.025]** | 158 (33.0) | 1.03 (0.76–1.40) [0.842] |
| | Missing | 13 (1.6) | 1 (7.7) | N/A | 6 (46.2) | N/A |
| Hypertension (N [%]) | No | 613 (76.1) | 143 (23.3) | Ref | 205 (33.4) | Ref |
| | Yes | 189 (23.4) | 31 (16.4) | **0.63 (0.41–0.97) [0.041]** | 57 (30.2) | 0.85 (0.60–1.21) [0.378] |
| | Missing | 4 (0.5) | 0 (0) | N/A | 3 (75.0) | N/A |
| Physically active (N [%]) | No | 134 (16.6) | 39 (29.1) | Ref | 41 (30.6) | Ref |
| | Yes | 672 (83.4) | 135 (20.1) | **0.61 (0.41–0.95) [0.024]** | 224 33.3) | 1.14 (0.77–1.72) [0.517] |
| Healthy diet (N [%]) | No | 16 (2.0) | 4 (25.0) | Ref | 6 (37.5) | Ref |
| | Yes | 788 (97.8) | 169 (21.4) | 0.82 (0.28–2.98) [0.741] | 258 (32.7) | 0.82 (0.30–2.42) [0.697] |
| | Missing | 2 (0.2) | 1 (50.0) | N/A | 1 (50.0) | N/A |
| Smoking (N [%]) | No | 628 (77.9) | | | 168 (26.8) | Ref |
| | Yes | 174 (21.6) | | | 97 (55.7) | **3.45 (2.44–4.89) [<0.001]** |
| | Missing | 4 (0.5) | | | 0 (0) | N/A |

(*Continued*)

**Table 1.** (Continued)

| Participant characteristics (Total participants = 806) | | | Physically active | | Healthy diet | |
|---|---|---|---|---|---|---|
| | | | N (%) | Unadjusted OR (95% CI) [p-value] | N (%) | Unadjusted OR (95% CI) [p-value] |
| Alcohol drinking (N [%]) | No<br>Yes<br>Missing | 538 (66.7)<br>265 (32.9)<br>3 (0.4) | 77 (14.3)<br>97 (36.6)<br>0 (0) | Ref<br>**3.45 (2.44–4.89) [<0.001]**<br>N/A | | |

CI = confidence interval, N = number, N/A = not applicable, OR = odds ratio, Ref = reference category, T2DM = type 2 diabetes mellitus. Missing data = 0 for each variable unless otherwise specified. Variables selected for multivariable regression analysis due to p≤0.2 are highlighted in bold.

*Excluded from regression analysis due to 100% of those with no education falling into the healthy diet group. Unadjusted ORs, 95% CIs and p-values calculated using univariate binomial logistic regression via the glm() function in R.

smoker (versus not a smoker) [OR 1.93, 95% CI 1.27–2.94], having a family history of T2DM (versus not) [OR 1.83, 95% CI 1.30–2.59], and being overweight (versus not overweight) [OR 1.48, 95% 1.01–2.18]. Several of these analyses are of borderline significance given the 95% CIs approach one (the threshold of no association). This includes the effect estimate for the association between being single/widowed/divorced and a healthy diet, between physical activity and smoking, between being a non-manual worker and alcohol drinking, and between being overweight and alcohol drinking.

In sensitivity analysis using WHtR ≥ 0.5 to define abdominal obesity (rather than the previously specified waist circumference thresholds), being an alcohol drinker was significantly associated with abdominal obesity in the adjusted analysis (OR 2.09, 95% CI 1.24–3.57) [Appendix 2: S1 File].

## Discussion

The prevalence of two 'positive' lifestyle behaviours amongst this cohort of people living with T2DM was high; 83.4% (95% CI 80.6–85.9%) of participants were categorised as being physically active and 97.8% (95% CI 96.5–98.7%) as having a healthy diet. This could be considered unexpected as both physical inactivity and an unhealthy diet are known risk factors for T2DM [24]. However, this cohort of people living with T2DM had received established diabetes clinic care including lifestyle advice. It is possible that some participants enacted lifestyle changes by increasing physical activity and healthy diet following diagnosis. Social desirability bias in the answering of the questionnaires on physical activity and diet is also possible. Alternatively, high levels of physical activity and a healthy diet may have preceded diagnosis despite the known risk factors for T2DM. A cross-sectional analysis amongst a national sample of Chinese people aged ≥45 years found the prevalence of physical inactivity was 19.3% (95% CI 18.3–20.4%), giving a converse physically active prevalence of approximately 80.7%; comparable to our estimate [25]. Comparable data on diet is difficult to find, however, our estimates of healthy diet prevalence appear to generally exceed those previously reported [26].

The prevalence of current smoking in this study was 21.6% (95% CI 18.8–24.6%), in keeping with the known high smoking prevalence in China [27,28]. The prevalence of alcohol drinking was 32.9% (95% CI 29.6–36.2%). Alcohol drinking is known to vary significantly across different geographical areas in China [29]. Previous Ningbo-specific estimates (based on the prevalence of being a monthly frequent alcohol drinker) were 52.5% for men, 12.3% for women, and 29.9% across both sexes [30]. Our estimate of alcohol drinking prevalence amongst this cohort of people living with T2DM is therefore likely comparable to the general population of Ningbo.

**Table 2. Factors associated with lifestyle behaviours assessed by multivariable regression analysis.**

| Participant characteristics | | Physically active* Adjusted OR (95% CI) [p-value] | Healthy diet[†] Adjusted OR (95% CI) [p-value] | Smoking[‡] Adjusted OR (95% CI) [p-value] | Alcohol drinking[§] Adjusted OR (95% CI) [p-value] |
|---|---|---|---|---|---|
| Age | 18–39 years | | Ref | Ref | Ref |
| | 40–59 years | | 1.48 (0.42–4.83) [0.527] | 0.70 (0.38–1.29) [0.258] | 1.75 (1.00–3.10) [0.051] |
| | ≥ 60 years | | 8.06 (0.82–193.00) [0.108] | 0.52 (0.21–1.25) [0.142] | 1.15 (0.55–2.42) [0.719] |
| Sex | Female | | | Ref | Ref |
| | Male | | | **41.1 (16.2–139.0) [0.001]** | **4.00 (2.62–6.20) [<0.001]** |
| Education level | No qualification | | | Ref | Ref |
| | Class 1–6 | | | 0.32 (0.10–1.04) [0.054] | 1.53 (0.70–3.53) [0.298] |
| | Class 7–12 | | | 0.59 (0.20–1.77) [0.328] | 1.19 (0.56–2.69) [0.670] |
| | College/university | | | **0.19 (0.06–0.65) [0.006]** | 1.13 (0.46–2.87) [0.974] |
| Occupation | Manual worker | Ref | Ref | Ref | Ref |
| | Non-manual worker | 0.85 (0.50–1.43) [0.541] | 1.29 (0.37–4.07) [0.669] | 1.12 (0.64–1.97) [0.695] | **1.69 (1.01–2.83) [0.045]** |
| | Never worked/retired | 1.11 (0.69–1.88) [0.693] | 2.29 (0.41–19.40) [0.381] | 0.55 (0.27–1.12) [0.099] | 1.24 (0.71–2.19) [0.445] |
| Residence | Urban | Ref | | | Ref |
| | Rural | **0.59 (0.39–0.88) [0.009]** | | | 0.81 (0.55–1.19) [0.290] |
| Marital status | Married | | Ref | Ref | Ref |
| | Single/widowed/divorced | | **0.23 (0.07–0.96) [0.029]** | **0.54 (0.09–0.80) [0.025]** | 0.48 (0.19–1.08) [0.091] |
| Health insurance | Yes | Ref | | | Ref |
| | No | 0.52 (0.24–1.19) [0.104] | | | 0.62 (0.23–1.51) [0.313] |
| Duration of T2DM | ≤1 years | | | Ref | Ref |
| | >1–5 years | | | 0.85 (0.49–1.49) [0.574] | 0.65 (0.40–1.05) [0.079] |
| | >5–10 years | | | 0.86 (0.47–1.58) [0.624] | 0.68 (0.40–1.15) [0.150] |
| | >10 years | | | 0.56 (0.31–1.00) [0.051] | 1.00 (0.62–1.63) [0.996] |
| Family history of T2DM | No | | | | Ref |
| | Yes | | | | **1.83 (1.30–2.59) [<0.001]** |
| Controlled T2DM | Yes | | Ref | | Ref |
| | No | | 0.43 (0.10–1.39) [0.201] | | 0.73 (0.51–1.04) [0.084] |
| Body mass index (BMI) | Not overweight | | | | Ref |
| | Overweight | | | | **1.48 (1.01–2.18) [0.048]** |
| | Obese | | | | 1.56 (0.95–2.59) [0.081] |
| Abdominal obesity | No | | | Ref | |
| | Yes | | | 1.13 (0.74–1.72) [0.575] | |
| Hypertension | No | Ref | | Ref | |
| | Yes | 1.42 (0.89–2.34) [0.158] | | 0.61 (0.36–1.00) [0.052] | |
| Physically active | No | | Ref | Ref | |
| | Yes | | 2.37 (0.72–6.91) [0.126] | **0.56 (0.33–0.96) [0.033]** | |
| Healthy diet | No | Ref | | | |
| | Yes | 2.14 (0.65–6.14) [0.176] | | | |
| Smoking | No | Ref | | | Ref |
| | Yes | 0.66 (0.42–1.03) [0.062] | | | **1.93 (1.27–2.94) [0.002]** |
| Alcohol drinking | No | | | Ref | |
| | Yes | | | **1.85 (1.22–2.82) [0.004]** | |

CI = confidence interval, N/A = not applicable, OR = odds ratio, Ref = reference category. Missing data excluded listwise.

*Data missing for 11/806. [†]Data missing for 22/806. [‡]Data missing for 4/806. [§]Data missing for 4/806. Significant results (p≤0.05) are highlighted in bold. Variables selected for multivariable analysis based on the threshold of p≤0.2 in univariate analysis. Adjusted ORs, 95% CIs and p-values calculated using multivariable binomial logistic regression via the glm() function in R.

The results were suggestive of an association between rural residence (versus urban) and reduced odds of being physically active (OR 0.59, 95% CI 0.39–0.88). A previous study amongst middle-aged and older Chinese people has found a similar association between urban residence and reduced odds of incident physical inactivity, though not prevalent physical inactivity [25]. This seems initially counter-intuitive, given the known association between

urbanisation and reduced physical activity [31]. Further, older cross-sectional survey data from China found a larger proportion of rural residents reported being physically active compared to urban residents (78.1% and 21.8%, respectively) [32,33]. Differences in measurement tools used to capture physical activity may account for some of the variations, however, these contradictions may be driven by wider complexities including an ageing population and health disparities with greater health disadvantages in rural areas that could predispose to physical inactivity [34]. Historically, most physical activity in China was undertaken in the work environment, with higher rates of manual labour in rural areas [33]. Rapid urbanisation may have resulted in a declining proportion of the rural population that is physically active, through impact on work patterns. Parallel rises in leisure-related physical activity in more economically developed areas may accentuate this pattern [35]. An alternative explanation may be the differential validity of the IPAQ questionnaire between the urban and rural participants of the study. For example, 25.3% of participants from urban areas had a college or university education, versus 6.2% in rural areas. Participants with fewer educational opportunities could have a reduced understanding of the required questionnaire responses. Research assessing physical activity patterns in urban and rural settings longitudinally may provide further clarity on the changes occurring in these settings.

Being single/widowed/divorced (versus married) was associated with reduced odds of having a healthy diet in the present study (OR 0.23, 95% CI 0.07–0.96). The effect estimate was suggestive of a 77% reduction though with wide 95% CIs consistent with a 93% reduction through to a 4% reduction. Given the CI approaching one, we would advise caution in the over-interpretation of this result. It is possible that dietary habits could be influenced by marital status, though this is likely to be a complex relationship with the potential for gender-based differences amongst other factors [36]. Qualitative research exploring the interplay between social relationships and diet in this setting may be of benefit.

Prominent associations found in this study were between male sex (versus female) and smoking (an estimated 41-fold increased odds), and male sex (versus female) and alcohol drinking (an estimated 4-fold increased odds). Caution is required in interpreting the association between sex and smoking status given there were very few women who were current smokers in the study, and thus the magnitude of the effect estimate may not be reliable. However, these findings are in keeping with the existing literature which demonstrates a higher prevalence of these harmful lifestyle behaviours among men versus women in China [27,30]. Smoking is known to increase the risk of diabetic complications [11,12]. Evidence for whether alcohol is an independent risk factor for T2DM is mixed and may interact with other factors such as gender and ethnicity [37,38]. However, alcohol consumption can increase the risk of hypoglycaemia and can negatively impact self-care behaviours [13,14]. The findings thus highlight the importance of addressing smoking and alcohol intake amongst men living with T2DM in Ningbo, given the high prevalence and interaction with T2DM.

The findings of our study are broadly supportive of the concept of lifestyle interventions that target multiple lifestyle behaviours simultaneously, given there is a correlation between several of the lifestyle behaviours. Being an alcohol drinker was associated with increased odds of being a smoker (and vice versa), whilst being physically active was associated with reduced odds of smoking, albeit with a 95% CI approaching one (OR 1.85, 95% CI 1.22–2.82; OR 1.93, 95% CI 1.27–2.94; and OR 0.56, 95% CI 0.33–0.96, respectively). Broad diabetes self-management education and support (DSME/S) programmes that address a wide variety of self-care behaviours simultaneously may be more efficient than trying to address each risk factor through separate efforts [39].

Other factors associated with reduced odds of smoking in the current study were having a college/university education (versus no qualification) and being single/widowed/divorced

(versus married) [OR 0.19, 95% CI 0.06–0.65; and OR 0.54, 95% CI 0.09–0.80, respectively]. For the former, this may reflect broader links between lower socio-economic status and higher smoking prevalence in China [27]. For the latter, data from a nationally representative survey in China was suggestive of an association between being single and lower smoking prevalence in men, which may account for our findings [40].

Other factors associated with increased odds of alcohol drinking were a family history of T2DM, being a non-manual worker (versus manual), and being overweight (versus not overweight) [OR 1.83, 95% CI 1.30–2.59; OR 1.69, 95% CI 1.01–2.83; and OR 1.48, 95% 1.01–2.18, respectively]. The latter two associations are of borderline significance. For the former (alcohol drinking and family history of T2DM), it is possible that less healthy behaviours could be more commonplace amongst families with higher rates of T2DM. In sensitivity analysis, using WHtR $\geq$ 0.5 to define abdominal obesity (rather than the previously specified waist circumference thresholds), being an alcohol drinker was significantly associated with abdominal obesity (adjusted OR 2.09, 95% CI 1.24–3.57). This may be a more sensitive measure of abdominal obesity compared to waist circumference thresholds which do not take height into account, and is an intuitive finding given the calorific content of alcohol [41].

To the authors' knowledge, this is the first study addressing the prevalence of lifestyle behaviours in a cohort of patients living with T2DM at a diabetes clinic in Ningbo, China; and may help to inform the development of future lifestyle behaviour support interventions at the clinic. Additional strengths of the study include generally low rates of missing data (Table 1), and an emphasis on the reliability of measurements through standardised processes, use of validated questionnaires, and duplicate anthropometric measurements.

Limitations of the study include the aforementioned risk of social desirability bias when participants answer questions about behaviours they perceive to have a value judgment. This may have restricted our ability to detect the true prevalence of 'negative' lifestyle behaviours. Measurement of lifestyle behaviours generally presents a challenge however future research in this area could make use of more objective methods [42,43]. The binary categorisation of the four lifestyle behaviours in the present study comes with some loss of information on the extent of lifestyle behaviours, however was necessary for the chosen analytical strategy. Data were not collected on comorbidities and complications which may also be associated with lifestyle behaviours. The cross-sectional nature of the study also prevented the establishment of cause and effect between the associated factors and the lifestyle behaviours. In addition, there were a high number of analyses in the present study which may increase the risk of false-positive associations. Those associations of borderline statistical significance should be interpreted with caution. Given the highly contextual nature of lifestyle behaviours, the generalisability of the findings outside of the Ningbo context or to those without T2DM is limited. Further, the clinic-based sampling strategy created potential selection bias, as the characteristics of participants attending the diabetes clinic may differ systematically from people living with T2DM more generally across Ningbo. Additionally, those from the clinic who opted to participate may differ from those who did not, and data on non-responders were not collected due to the large workload in the busy clinic setting of the study. It could be anticipated that people living with T2DM who were engaged and satisfied with clinic care and willing to take part in a study [44] may be more likely to also engage with healthy lifestyle advice, which could have resulted in an overestimate of healthy lifestyle behaviours in our sample. However, the study provides useful information on a specified population, helping to identify potential areas for targeted lifestyle interventions. Complementary qualitative research which explores the lived experiences of those with T2DM regarding their lifestyle behaviours would help elucidate barriers and facilitators of lifestyle change in this context.

## Conclusions

This study has demonstrated the prevalence of four key lifestyle behaviours amongst people living with T2DM attending a diabetes clinic in Ningbo, China. We found high levels of a healthy diet. Levels of physical activity, smoking, and alcohol drinking appeared in keeping with those of the general population. Marked associations were demonstrated between male sex and smoking (41-fold increased odds) and male sex and alcohol drinking (4-fold increased odds). Correlation between lifestyle factors was demonstrated including between alcohol drinking and smoking and between physical activity and reduced smoking. General DSME/S programmes that address multiple lifestyle-related risk factors simultaneously may be beneficial in this population. Specific interventions targeting smoking cessation and reduction in alcohol drinking may be of benefit to men living with T2DM attending a diabetes clinic in Ningbo.

## Supporting information

**S1 Checklist. Inclusivity in global research.** PLOS ONE Checklist.
(DOCX)

**S1 File. Sensitivity analysis.** Sensitivity analysis using waist-to-height ratio (WHtR) to define abdominal obesity instead of waist circumference thresholds.
(DOCX)

**S2 File. Dataset.** Dataset supporting the findings.
(XLSX)

## Acknowledgments

The authors thank the study participants.

## Author Contributions

**Conceptualization:** Jialin Li, Li Li, Kaushik Chattopadhyay.

**Data curation:** Jialin Li.

**Formal analysis:** Naomi Carter, Kaushik Chattopadhyay.

**Funding acquisition:** Jialin Li, Li Li, Kaushik Chattopadhyay.

**Investigation:** Jialin Li, Li Li, Kaushik Chattopadhyay.

**Methodology:** Naomi Carter, Jialin Li, Li Li, Kaushik Chattopadhyay.

**Project administration:** Jialin Li, Miao Xu, Li Li, Shengnan Xu, Xuelan Fan, Shuyan Zhu, Kaushik Chattopadhyay.

**Resources:** Jialin Li, Miao Xu, Li Li, Shengnan Xu, Xuelan Fan, Shuyan Zhu, Kaushik Chattopadhyay.

**Supervision:** Jialin Li, Miao Xu, Li Li, Prit Chahal, Kaushik Chattopadhyay.

**Visualization:** Naomi Carter, Kaushik Chattopadhyay.

**Writing – original draft:** Naomi Carter, Kaushik Chattopadhyay.

**Writing – review & editing:** Naomi Carter, Jialin Li, Miao Xu, Li Li, Shengnan Xu, Xuelan Fan, Shuyan Zhu, Prit Chahal, Kaushik Chattopadhyay.

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
