## [Decision Letter · Decision Letter 0]

1 Aug 2023

PONE-D-23-19169Lifestyle behaviours and associated factors among people with type 2 diabetes in Ningbo, China: A cross-sectional studyPLOS ONE

Dear Dr. Chattopadhyay,

Thank you for submitting your manuscript to PLOS ONE. After careful consideration, we feel that it has merit but does not fully meet PLOS ONE’s publication criteria as it currently stands. Therefore, we invite you to submit a revised version of the manuscript that addresses the points raised during the review process.

We look forward to receiving your revised manuscript.

Kind regards,

Hidetaka Hamasaki

Academic Editor

PLOS ONE

2. In the ethics statement in the Methods, you have specified that verbal consent was obtained. Please provide additional details regarding how this consent was documented and witnessed, and state whether this was approved by the IRB"

3. Please include a complete copy of PLOS’ questionnaire on inclusivity in global research in your revised manuscript. Our policy for research in this area aims to improve transparency in the reporting of research performed outside of researchers’ own country or community. The policy applies to researchers who have travelled to a different country to conduct research, research with Indigenous populations or their lands, and research on cultural artefacts. The questionnaire can also be requested at the journal’s discretion for any other submissions, even if these conditions are not met.  Please find more information on the policy and a link to download a blank copy of the questionnaire here: https://journals.plos.org/plosone/s/best-practices-in-research-reporting. Please upload a completed version of your questionnaire as Supporting Information when you resubmit your manuscript.

Additional Editor Comments:

Dear authors,

Thank you for submitting your valuable work to PLOS ONE. Your research appears to have been ethically conducted and scientifically valid; however, the reviewers require you to make major revisions.

In addition to the reviewers' comments, the editor thinks that the authors should describe the sampling method used in this study and discuss potential biases in the manuscript.

I look forward to reading your revised manuscript soon.

Sincerely,

Reviewers' comments:

Reviewer's Responses to Questions

**Comments to the Author**

1. Is the manuscript technically sound, and do the data support the conclusions?

Reviewer #1: Partly

Reviewer #2: Partly

2. Has the statistical analysis been performed appropriately and rigorously? 

Reviewer #1: N/A

Reviewer #2: Yes

3. Have the authors made all data underlying the findings in their manuscript fully available?

Reviewer #1: Yes

Reviewer #2: Yes

4. Is the manuscript presented in an intelligible fashion and written in standard English?

Reviewer #1: Yes

Reviewer #2: Yes

5. Review Comments to the Author

Reviewer #1: The results of the present study may be meaningful for the study participants. However, I have some comments on the study.

Line 66-68: The study participants were selected from single hospital, which indicated that the study participants can not represent the status of the total T2DM patients living in Ningbo. In other words, I considered that there may be serious and uncontrolled representativeness bias in the present study.

Line 74: The diagnostic criteria of T2DM should be described in the text. Is there difference between diagnostic criteria of T2DM of China and of IDF / ADA? If there is, then why?

Line 88-89: The authors should clearly describe if there were study participants whose two readings were significantly different, and the significant difference should also be defined. If there were, then the measurement may be untrusted and that patient should be excluded.

Line 101-103, table 1 and 2: I suggest that waist-to-height ratio (WHtR) should be used to define abdominal obesity because waist circumference alone did not take differences in height into account. At least, the associations of WHtR with lifestyle behaviours and associated factors should be researched in the present manuscript.

Line 136: The study should include information on duration of smoking cessation in those ex-smokers. Actually, the years immediately following smoking cessation may represent a time of increased risk for DM (PMID: 34964876).

Line 207-208: It is difficult to exactly assess the intensity of physical activity (PA). The present study obtained PA data using IPAQ questionnaire (Line 109). I guess there may be another reasons: for example, the authors should describe if there was difference in education level between rural and urban participants in the results section. If education level of rural participants was lower than their urban counterparts, than the reliability of their IPAQ questionnaire results may be lower because of lower ability to understand and/or patience.

Line 233-241: The association between alcohol drinking and DM should be discussed in more detail. Moderate alcohol intake had minimal detrimental effects on long-term blood glucose management (PMID: 26294775). Furthermore, though the authors of a recent published study (PMID: 37414658) speculated that the emerged discomfort reduced alcohol consumption frequency in DM population of China, that frequency was not significantly associated with DM risk.

Line 264-265: Though that may be the first study “in a cohort of patients living with T2DM in Ningbo”, I still considered that the innovativeness of the present study is quite limited. As the authors mentioned, Ningbo ist an economically developed city in the eastern, coastal province of Chekiang (Line 47), however, I don't understand why the authors selected Ningbo as the studied city. The status of the city can not represent the most developed regions of China, and that city may not even represent the province of Chekiang. Furthermore, as above-mentioned, the study participants of the present study can not represent the status of the total T2DM patients living in Ningbo. Therefore, I considered that the present study may be only beneficial to those 806 study participants, and other T2DM patients may not benefit from the present study. The authors should clearly explain the innovativeness of the present study and the representativeness of the sample.

Line 269-281: There were some other limitations that the authors should mention in the text. For example, the cause and effect relationship can not be examined in the present cross-sectional study.

Line 276-278: I still considered that the present study only provides information in those 806 study participants because they can not represent the status of the total T2DM patients living in Ningbo.

Line 284-285: Again, the prevalence found in the present study was not the prevalence amongst T2DM patients in Ningbo, but just the prevalence amongst those 806 study participants.

Reviewer #2: This paper presents a cross sectional study on a descriptive analyses of life style risk factors and their associated factors in Ningbo, China. I have some concerns regarding selection bias due to the unexpectedly high prevalence of healthy diet and physical activity among the sample of individuals with type 2 diabetes, as this disease is strongly a lifestyle dependent disease. The authors attempted to discuss the unexpected result, but the aspect of selection bias was missing in the discussion and the limitations.

- How did you address selection bias?

- How were the participants asked to join the study? Were there any incentives?

- How was the response rate? It could very well be that those who accepted to participate are the ones who are more aware of their health. This is especially relevant because anthropometric measurements were taken, which would be uncomfortable for overweight or obese individuals.

- Is it possible to perform a non-responder analysis? This might also help understand the results of the study

- Did the questionnaire also include question on complications and comorbidities? If no, why, and if yes, why were they not considered in the analysis as they are highly relevant (for example by stratifying the sample by morbidity level)?

- Why were the extent of smoking and alcohol (for example number of cigarettes per day or frequency of alcohol intake) not considered?

The study would have a potential for being worth publishing, but only after addressing the issue of selection bias by adding justifications for example, and by including the response rate and reflecting on it.

6. PLOS authors have the option to publish the peer review history of their article (what does this mean?). If published, this will include your full peer review and any attached files.

Reviewer #1: No

Reviewer #2: No

---

## [Author Response · Author response to Decision Letter 0]

10 Aug 2023

Please find attached the letter. Thank you.

---

## [Decision Letter · Decision Letter 1]

30 Aug 2023

PONE-D-23-19169R1Lifestyle behaviours and associated factors among people with type 2 diabetes attending a diabetes clinic in Ningbo, China: A cross-sectional studyPLOS ONE

Dear Dr. Chattopadhyay,

Thank you for submitting your manuscript to PLOS ONE. After careful consideration, we feel that it has merit but does not fully meet PLOS ONE’s publication criteria as it currently stands. Therefore, we invite you to submit a revised version of the manuscript that addresses the points raised during the review process.

ACADEMIC EDITOR:Thank you for submitting the revised version of your manuscript.Reviewer#2 has recommended rejecting your manuscript; however, the editor believes that the authors have appropriately addressed the issue regarding the sampling bias and mentioned its limitation in the revised manuscript. Therefore, the editor requires you to make minor revisions based on Reviewer#1's comments.I look forward to reading your refined manuscript.  

We look forward to receiving your revised manuscript.

Kind regards,

Hidetaka Hamasaki

Academic Editor

PLOS ONE

Journal Requirements:

Reviewers' comments:

Reviewer's Responses to Questions

**Comments to the Author**

1. If the authors have adequately addressed your comments raised in a previous round of review and you feel that this manuscript is now acceptable for publication, you may indicate that here to bypass the “Comments to the Author” section, enter your conflict of interest statement in the “Confidential to Editor” section, and submit your "Accept" recommendation.

Reviewer #1: (No Response)

Reviewer #2: (No Response)

2. Is the manuscript technically sound, and do the data support the conclusions?

Reviewer #1: Yes

Reviewer #2: (No Response)

3. Has the statistical analysis been performed appropriately and rigorously? 

Reviewer #1: N/A

Reviewer #2: Yes

4. Have the authors made all data underlying the findings in their manuscript fully available?

Reviewer #1: Yes

Reviewer #2: Yes

5. Is the manuscript presented in an intelligible fashion and written in standard English?

Reviewer #1: Yes

Reviewer #2: Yes

6. Review Comments to the Author

Reviewer #1: Most answers were reasonable. However, I considered that there are still some comments which the authors should response.

Comment #3: Lines 140-142: The authors should clearly define the implausible value in the revised manuscript.

Comment #7: The pages of reference #38 were volume 33 issue 9 1693-1701, but not S0939-4753(23)00091-1.

Comment #8: I agree that the present findings worth reporting and sharing after the phrase changed to “people living with T2DM attending a diabetes clinic…”. I suggest the authors provide more information on that diabetes clinic (i.e. the first affiliated hospital of Ningpo University), to explain why that clinic, but not other clinic, was chosen as the studied clinic. For example, if the first affiliated hospital had the best medical and diagnostic facilities for T2DM in Ningpo, or not?

Moreover, there were some other suggestions. First, the authors should carefully check the content of manuscript to avoid minor errors, for example, the above-mentioned reference #38 and the reference order. Second, it would be better if the DOI of all references were listed.

Reviewer #2: Even though the authors did a good job in the analyses and writing the manuscript, the study appears to have serious potential selection bias that was not addressed properly during the sampling phase. Data essential for calculating response rate and performing a basic non-responder analysis is not available, and this cannot be neglected especially that as the authors also explain, the sample is not representative and the generalisability of the data is very limited. In fact, I also doubt the generalisability of the data to the specific diabetes centre in Ningbo where the sample was recruited, since no information what so ever exists on non-responders. Therefore, I have problems in trusting the results of the study. In addition, no convincing justification is provided as to why the study was done using this specific diabetes centre (a collaborative partnership with the hospital is not a sufficient justification), so that I am not pretty sure what the results of this study would add to the literature, or whether the results are worth publishing. Therefore, I would unfortunately not recommend accepting the paper for publication.

7. PLOS authors have the option to publish the peer review history of their article (what does this mean?). If published, this will include your full peer review and any attached files.

Reviewer #1: No

Reviewer #2: No

---

## [Author Response · Author response to Decision Letter 1]

5 Sep 2023

Monday, 4th September 2023

Dear Hidetaka Hamasaki,

Re: PONE-D-23-19169R1

Lifestyle behaviours and associated factors among people with type 2 diabetes in Ningbo, China: A cross-sectional study

Many thanks for your further correspondence regarding the above manuscript following the second round of peer review, and requesting minor revisions based on reviewer #1’s comments. Please see point-by-point responses below. 

Yours Sincerely,

Dr Kaushik Chattopadhyay, on behalf of the authors

 

Minor revisions following second round of peer review – point-by-point response.

Editor:

1) Journal Requirements:

The in-text citations and reference list have been reviewed, with the following corrections. In text citations were in curved brackets e.g. (1) which have now been changed to square brackets e.g. [1] as per PLOS One submission guidelines. The reference in line 100 has been corrected from (17) to [18]. The reference in line 123 has been corrected from (20) to [21]. DOIs have been added to the reference list where available. Publication months have been added to the reference list where available. Page numbers for reference 38 were corrected. The full updated reference list with tracked-changes is available from lines 356 to 512. To the authors’ knowledge, no retracted articles have been cited.

Reviewer 1:

1) Comment #3: Lines 140-142: The authors should clearly define the implausible value in the revised manuscript.

The decision as to whether a value was implausible was at the discretion of the researchers during the data cleaning process. A record was kept of these decisions. The manuscript text has been changed to reflect this as below (lines 139-140):

‘During data cleaning, data were screened for implausible values (as judged by the researchers), and these were deleted where found and treated as missing.’

2) Comment #7: The pages of reference #38 were volume 33 issue 9 1693-1701, but not S0939-4753(23)00091-1.

Thank you, this has been corrected (please see reference 38).

3) Comment #8: I agree that the present findings worth reporting and sharing after the phrase changed to "people living with T2DM attending a diabetes clinic...". I suggest the authors provide more information on that diabetes clinic (i.e. the first affiliated hospital of Ningpo University), to explain why that clinic, but not other clinic, was chosen as the studied clinic. For example, if the first affiliated hospital had the best medical and diagnostic facilities for T2DM in Ningpo, or not?

An additional line has been added with further information about the First Affiliated Hospital of Ningbo University, as below (lines 72-73). We are not able to draw comparisons regarding quality of care across T2DM care providers in Ningbo in the context of this research paper. 

As well as providing specialist health and medical services, the First Affiliated Hospital of Ningbo University is involved in medical research and education (6).

4) Moreover, there were some other suggestions. First, the authors should carefully check the content of manuscript to avoid minor errors, for example, the above-mentioned reference #38 and the reference order.

The manuscript has been further proof-read for correction of minor errors. The sentence in line 21-23 was reordered to improve readability. Line 50: the word ‘being’ was removed as it was unnecessary. 

As above, the in-text citations and reference list have been reviewed, with the following corrections. In text citations were in curved brackets e.g. (1) which have now been changed to square brackets e.g. [1] as per PLOS One submission guidelines. The reference in line 100 has been corrected from (17) to [18]. The reference in line 123 has been corrected from (20) to [21]. DOIs have been added to the reference list where available. Publication months as well as the year have been added where available. Page numbers for reference 38 were corrected. The full updated reference list with tracked-changes is available from lines 356 to 512. To the authors’ knowledge, no retracted articles have been cited.

5) Second, it would be better if the DOI of all references were listed.

Many thanks, DOIs have been added where available (please see reference list).

---

## [Decision Letter · Decision Letter 2]

17 Oct 2023

PONE-D-23-19169R2Lifestyle behaviours and associated factors among people with type 2 diabetes attending a diabetes clinic in Ningbo, China: A cross-sectional studyPLOS ONE

Dear Dr. Chattopadhyay,

Thank you for submitting your manuscript to PLOS ONE. After careful consideration, we feel that it has merit but does not fully meet PLOS ONE’s publication criteria as it currently stands. Therefore, we invite you to submit a revised version of the manuscript that addresses the points raised during the review process.

We look forward to receiving your revised manuscript.

Kind regards,

Hidetaka Hamasaki

Academic Editor

PLOS ONE

Journal Requirements:

Reviewers' comments:

Reviewer's Responses to Questions

**Comments to the Author**

1. If the authors have adequately addressed your comments raised in a previous round of review and you feel that this manuscript is now acceptable for publication, you may indicate that here to bypass the “Comments to the Author” section, enter your conflict of interest statement in the “Confidential to Editor” section, and submit your "Accept" recommendation.

Reviewer #1: (No Response)

Reviewer #3: (No Response)

2. Is the manuscript technically sound, and do the data support the conclusions?

Reviewer #1: (No Response)

Reviewer #3: Yes

3. Has the statistical analysis been performed appropriately and rigorously? 

Reviewer #1: (No Response)

Reviewer #3: Yes

4. Have the authors made all data underlying the findings in their manuscript fully available?

Reviewer #1: (No Response)

Reviewer #3: No

5. Is the manuscript presented in an intelligible fashion and written in standard English?

Reviewer #1: (No Response)

Reviewer #3: Yes

6. Review Comments to the Author

Reviewer #1: Lines 72-73: I guess that the authors may misunderstand my previous suggestion to provide more information on the diabetes clinic. The purpose of that suggestion is to clarify the representativeness of the present sample. If the present sample can not represent those T2DM patients living in Ningpo, or even not represent all T2DM patients attending that diabetes clinic, then I considered that the present study may be suitable for publication in some other forms, but not in the form of Research Article, though the findings worth sharing.

If possible, I suggest inviting a statistical editor to assess whether the present sample can represent the studied population. I have no more comments on the present manuscript and I considered that it may be more suitable if the decision (accept or reject) was made by that statistical editor.

Reviewer #3: Here is a list of specific comments. Note: line and page numbering in reviews and comments is based on ruler applied in Editorial Manager-generated PDF.

1. Page 7, line 143: “Researchers” seemed vague. I suggest listing initials of those responsible for the data cleaning to provide credibility.

2. Page 7, line 149: I suggest including the R package where the PropCIs function comes from.

3. Page 7, line 150: I suggest replacing “multivariate” with ‘multivariable.’

4. Page 7, lines 157–158: Why can’t the waist-to-height ratio be considered as one of the covariate? It did not seem necessary to include it as a sensitivity analysis.

5. Page 17, lines 261–262: I suggest paying less attention on the odds ratio of gender and smoking. There were too few females who were current smokers. Therefore, the estimates might not be reliable.

7. PLOS authors have the option to publish the peer review history of their article (what does this mean?). If published, this will include your full peer review and any attached files.

Reviewer #1: No

Reviewer #3: No

---

## [Author Response · Author response to Decision Letter 2]

26 Oct 2023

Wednesday, 25th October 2023

Dear Hidetaka Hamasaki,

Re: PONE-D-23-19169R1

Lifestyle behaviours and associated factors among people with type 2 diabetes in Ningbo, China: A cross-sectional study

Many thanks for your further correspondence regarding the above manuscript following statistician review. Please see point-by-point responses below. 

Yours Sincerely,

Dr Kaushik Chattopadhyay, on behalf of the authors

 

Minor revisions following statistician peer review – point-by-point response.

Reviewer 1

Comment #1:

Lines 72-73: I guess that the authors may misunderstand my previous suggestion to provide more information on the diabetes clinic. The purpose of that suggestion is to clarify the representativeness of the present sample. If the present sample can not represent those T2DM patients living in Ningpo, or even not represent all T2DM patients attending that diabetes clinic, then I considered that the present study may be suitable for publication in some other forms, but not in the form of Research Article, though the findings worth sharing.

If possible, I suggest inviting a statistical editor to assess whether the present sample can represent the studied population. I have no more comments on the present manuscript and I considered that it may be more suitable if the decision (accept or reject) was made by that statistical editor.

Many thanks for the suggestion of inviting a statistical editor to review our manuscript; comments from the statistician are addressed below.

Reviewer 3

Comment #1:

Page 7, line 143: "Researchers" seemed vague. I suggest listing initials of those responsible for the data cleaning to provide credibility.

Amended as suggested in lines 141-143:

‘During data cleaning, data were screened for implausible values (data cleaning completed by NC) , and these were deleted where found and treated as missing.’

Comment #2:

Page 7, line 149: I suggest including the R package where the PropCIs function comes from.

Apologies, PropCI is the package name not the function name. Corrected as below on line 149:

‘Clopper-Pearson 95% confidence intervals (CIs) calculated using the PropCIs package in R.’

Comment #3:

Page 7, line 150: I suggest replacing "multivariate" with 'multivariable.'

Amended throughout as suggested.

Comment #4:

Page 7, lines 157–158: Why can't the waist-to-height ratio be considered as one of the covariate? It did not seem necessary to include it as a sensitivity analysis.

We defined abdominal obesity based on reference ranges specific to the Chinese population (lines 107-108 of the manuscript) rather than using waist-to-height ratio. A previous reviewer comment suggested we could use waist-to-height ratio as an alternative way to define abdominal obesity. To avoid post-hoc changes to the analytical strategy we included this as a sensitivity analysis rather than repeating all analyses using a new definition of abdominal obesity.

Comment #5:

Page 17, lines 261–262: I suggest paying less attention on the odds ratio of gender and smoking. There were too few females who were current smokers. Therefore, the estimates might not be reliable.

Amended taking this into account, as below (lines 260-265):

‘Prominent associations found in this study were between male sex (versus female) and smoking (an estimated 41-fold increased odds), and male sex (versus female) and alcohol drinking (an estimated 4-fold increased odds). Caution is required in interpreting the association between sex and smoking status given there were very few women who were current smokers in the study, and thus the magnitude of the effect estimate may not be reliable. However, these findings are in keeping with the existing literature…’

---

## [Editor Report · Decision Letter 3]

27 Oct 2023

Lifestyle behaviours and associated factors among people with type 2 diabetes attending a diabetes clinic in Ningbo, China: A cross-sectional study

PONE-D-23-19169R3

Dear Dr. Chattopadhyay,

We’re pleased to inform you that your manuscript has been judged scientifically suitable for publication and will be formally accepted for publication once it meets all outstanding technical requirements.

Kind regards,

Hidetaka Hamasaki

Academic Editor

PLOS ONE

Additional Editor Comments (optional):

Kudos to the authors for revising their work. Thank you for your effort. I believe that you have adequately addressed the issues raised by the reviewers. Celebrate this achievement, and good luck with your next endeavor!"
---

## [Editor Report · Acceptance letter]

9 Nov 2023

PONE-D-23-19169R3 

Lifestyle behaviours and associated factors among people with type 2 diabetes attending a diabetes clinic in Ningbo, China: A cross-sectional study 

Dear Dr. Chattopadhyay:

I'm pleased to inform you that your manuscript has been deemed suitable for publication in PLOS ONE. Congratulations! Your manuscript is now with our production department. 

Kind regards, 

on behalf of

Dr. Hidetaka Hamasaki 

Academic Editor

PLOS ONE